# Backyard Biomes: Is Anyone There? Improving Public Awareness of Urban Wildlife Activity

Loren L. Fardell [1,*] , Chris R. Pavey [2] and Chris R. Dickman [1]

1   School of Life and Environmental Sciences, University of Sydney, Camperdown, NSW 2006, Australia; chris.dickman@sydney.edu.au
2   CSIRO Land and Water, Winnellie, NT 0822, Australia; chris.pavey@csiro.au
*   Correspondence: loren.fardell@gmail.com

**Abstract:** Wildlife are increasingly being found in urban habitats, and likely rely on some resources in suburban household yards, which exposes them to the effects of yard management and human and pet activities. We compared the relationships between these potential disturbances and benefits to the number of different types of wildlife sighted by householders, using written surveys. Owing to the inability of many household respondents to identify animals to the species or genus level, each different 'type' of animal individually listed was counted to generate the total number of types of wildlife observed by each household. We found that relatively more types of wildlife were observed by residents whose yards provided ease of faunal access under or through fences, had reduced pesticide use, increased levels of anthropogenic noise, and increased presence of pets in yards. The latter two associations likely relate to the increased opportunities to observe wildlife in yards that each creates. We also investigated the use of yards by wildlife and domestic pets in open compared to more vegetated habitats by day and night, using motion-sensor cameras. All animals observed were compared to the activity of introduced brown and black rats (*Rattus norvegicus*, *R. rattus*), owing to their wild origins but long commensal history with humans. Camera images indicated that animals' natural activity periods were maintained in yards. Brown antechinuses (*Antechinus stuartii*), northern brown bandicoots (*Isoodon macrourus*), domestic cats (*Felis catus*) and native birds (species as listed below) each preferred sheltered or vegetated habitats over open habitats, when compared to the introduced rats that showed little habitat preference. However, unlike the other species, the native birds used open areas more than vegetated or sheltered areas when compared within their group only. The common brushtail possum (*Trichosurus vulpecula*) was observed to use open areas comparatively more than the introduced rats, but used vegetated or sheltered habitats more when compared to self only. The domestic dog (*Canis familiaris*) and red fox (*Vulpes vulpes*) used open areas more than vegetated or sheltered areas, when compared to the introduced rats, and against themselves. This indicated a level of coping with urban stressors by the native animals, but with a reliance on more vegetated habitats to allow for natural stress-relieving behaviours of escape or hiding. Here, we offer insights into how each of these findings may be used to help educate and motivate increased household responsibility for urban wildlife conservation.

**Keywords:** wildlife-friendly gardening; urban wildlife biodiversity; urban conservation management; social ecology; household yard

## 1. Introduction

As habitat modification [1] and climate change [2] intensify globally, urban environments are becoming the best or only available habitats for fauna [3], with wildlife increasingly being found to reside in them [4–8]. Wildlife diversity and threatened species may nonetheless be under pressure from rapid urbanisation, especially in remnant habitats proximal to urban areas [1,9,10]. Some species may thrive under urban conditions given

the benefits of access to reliable supplementary resources, including water and food, provided by human activities [11–15]. The 'human shield' effect, in creating a buffer from larger predators for prey [16], and/or supplementary food resources that reduce predation pressure on smaller animals, can also benefit small prey species in urban habitats [17,18]. However, some fauna species may not benefit from such rewards or are unable to endure the additional risks and stressors that human activities create in urbanised environments, such as pesticides [19], domestic pets [20,21], and introduced light and noise [22]. For such species, these negative impacts can result in local extinctions, in turn reducing local wildlife diversity [11].

In patchy urban areas, household yards or gardens connect to green spaces and larger nature or conservation reserves, creating a mosaic of resource-abundant islands amid urban development [23,24]. Despite their small size [8], yards can help to support native animals, particularly if they provide adequate resources, habitats, and refuges [25]. Based on 'landscapes of fear' theory, where prey perceive and respond to the predation risk associated with particular habitat components [26], yards that are more open and offer little shelter or canopy cover, with minimal vegetation, may be perceived as risky due to their exposure to human disturbances [27,28], whereas yards with cover from more vegetation or refuge structures may be perceived as less stressful [29]. Domestic pets in yards, being mostly predators [20,21] or large-bodied species, and human activity, which has been likened to exerting predator-like pressures [30,31], can therefore each directly influence wildlife activity in yards. Under such pressures, wildlife may change their active periods to avoid interactions with these stressors, and/or to gain better access to resources [32,33]. Animals that have successfully moved into human-dominated habitats include introduced rodents such as the brown rat (*Rattus norvegicus*) and black rat (*R. rattus*), which persist despite people's constant efforts to remove them [34,35]. These commensal introduced rats thus exhibit some degree of successful management of the stressors associated with being close to humans, and may be used in comparison to other species that are less accustomed to human activity to understand if they are coping with the stressors.

Given the resource potential, connectedness, and collective area of household yards, conservation of wildlife in urban habitats requires an equal understanding of the use of green spaces and surrounding nature or conservation reserves, as well as of household yards and activities that may act as stressors that negatively influence wildlife activity [36]. It follows then that people's perceptions towards sharing their space with wildlife are also important to consider. The actions of many individuals can contribute to biodiversity loss in urban environments and in remnant environments proximal to them; as such, urban conservation faces sociological challenges as well as ecological ones [37]. A failure to include the perspectives of people may reduce the success of urban wildlife conservation, as people may find directives unacceptable and be unwilling to support them if they have not been engaged [38,39].

Whilst there may be rules governing land management by local councils to consider utilities and landscape laws, including the preservation or removal of certain trees, the management of household yards and gardens is largely outside of government control. By-laws and local government regulations may encourage wildlife-friendly yards, which may be supported by non-government organisations [9], but usually these are not enforced. The success of wildlife conservation initiatives, therefore, relies on people understanding, and directly responding and reducing activities that negatively affect local wildlife [40]. However, reduced interaction with nature may foster a reduced connection to it, and hence reduce the willingness of people to implement conservation activities that they perceive will affect their everyday life [41,42]. A lack of understanding of how yards can help in sustaining urban wildlife [43] may also reduce people's ability to form connections and develop a level of conservation responsibility in how yards can be maintained and potential disturbances reduced [41]. Therefore, as households need to be engaged to practice wildlife-friendly gardening, in order to maximise urban wildlife conservation [9], the challenge lies in motivating such actions.

Providing background information to change conservation behaviour is often not effective on its own [44,45], but when paired with opportunities to connect people with nature such as via volunteering, citizen science, or immersive educational experiences, it can lead to beneficial behavioural changes [46–48]. Another effective driver of behavioural change is the level of personal and collective competence—being part of a wider community or group can effect change and help to achieve goals [42]. Human behavioural changes, in respect to household yard activities, are constrained by social norms, as well as attitudes/identity/interests, routines/habits, and personal capability restrictions [49,50]. The size of a yard and its function, e.g., whether it contains fruit or vegetable plants, also affect how a yard is managed and attitudes towards it [50,51]. Understanding people's connection to nature and if they encounter wildlife in their yards will also help to determine people's motivations to implement wildlife conservation practices. Attracting visually attractive wildlife to yards, such as birds or butterflies, can be a useful initial motivator to manage household yards for conservation [39,48]. Once a positive attitude has developed, it may then be extended to other animals so that yards can be better managed to support them.

Although urban biodiversity research is increasing [52,53], information on household yard activity by wildlife remains limited, despite evident interest by many people in how best to conserve it [23,54]. To better understand wildlife use of household yards and peoples' attitudes and activities that may influence wildlife, as resources or stressors, we distributed a written survey to residents within walking distance of a large conservation area in a patchy urban environment in eastern Australia, and placed motion-sensor cameras in the yards of volunteers. In doing so, we investigated: (1) if there is a relationship between wildlife observed by households and potentially influential human activities that provide resources or stressors; and (2) if native wildlife prefer open or vegetated/covered areas in household yards and if they shift their active time, relative to that of the successfully commensal introduced rats. We predicted that resources and disturbances would, respectively, attract and deter wildlife to household yards, and that wildlife that were coping with human-introduced stressors would use yards in a similar fashion to introduced rats.

## 2. Materials and Methods

### 2.1. Study Area

The study area encompasses the suburbs of Whitebridge and Dudley, two connected medium–low-density urban areas within the Lake Macquarie district of New South Wales, Australia. They are bordered by the 534 ha Glenrock State Conservation Area (GSCA), and are interspersed with numerous green spaces that include large plots of empty state-owned land, and have corridors that connect the nature reserves to the green spaces and to the larger GSCA [55,56]. Many of these areas, irrespective of size, contain remnant vegetation that pre-dates European disturbance [55,56]. Communities of small native animals occupy both the green spaces and the larger GSCA, as do two species of introduced mesopredator: the red fox (*Vulpes vulpes*) and domestic cat (*Felis catus*) [27,28,55]. Further description and study area maps are provided in our previous publication [57].

### 2.2. Household Activity Surveys

This project was conducted under animal ethics (2017/1275) and human ethics (2017/977) approval from the University of Sydney, and under a New South Wales Scientific License (SL102024). A printed written survey (see Supplemental Information) was distributed via letterbox drop to 400 houses within the Whitebridge and Dudley areas. The properties were selected if they were within walking distance of the GSCA and at least one green space that connected to it. The survey posed questions about householders' actions that may affect wildlife activity, including the number of pets, livestock, and poultry owned, their frequency of access to the household yards and off-property areas; the type of yard relative to levels of modification and naturalness; the frequency of garden watering and presence of a permanent open water source in the yard; the frequency of application of fertilisers and pesticides, and the types used; the frequency of use of light in yards, and the wattage of the light; the frequency of

anthropogenic noises from music or appliances that could be heard in the yard; the presence of gaps in fencing and the fence material; wildlife activity or signs of activity in yards; and the frequency and type of activities carried out by people in the nearby conservation area. Responses and written consent were returned via email or through use of a stamped envelope that we provided.

Answers were open ended without scaled options to get a better understanding of householders' yard structures and activities without the possible bias from pre-defined categories. Responses were grouped according to frequency, activity, or type. To classify yard type, we used descriptions given by respondents about modifications and naturalness of their yards and created five categories. Modifications included any human-made structures or alterations to the landscape as it naturally occurred, including garden beds and cultivation of introduced flora. Naturalness included the landscape as it naturally occurred, including cultivating flora that naturally occurred within the surrounding area. The categories were ordered in increasing modification and decreasing naturalness. *Semi-natural* yards contained vegetation that was native in the area, were connected to a surrounding nature reserve, had patches of open grass, few structures, and few exotic plants. If a yard bordered a nature reserve but had large areas of cleared open grass, few remnant native trees, numerous introduced plants, some structures including fencing and a pool, it was classified as *modified-natural*. If a yard had largely open grass, garden beds, a shed or chicken pen, and a mix of exotic and native trees and shrubs, then it was classified as *semi-modified*. If a yard had a combination of shed, pool and fence, multiple garden beds, mostly open grass and few trees or shrubs, mostly exotic, it was classified as *modified*. If a yard had the same as listed in modified but with more structures, very limited grass, mostly pavers or cement, and only potted or few garden bed plants that were mostly exotic, then it was classified as *very modified*. Fences were grouped as being made of open or fully enclosed material in combination with whether or not there was a gap between the fence and ground, so that there were four categories: *open gap*, *full gap*, *open ground*, and *full ground*. Wildlife activity or signs of activity in yards were counted as different 'types' of wildlife owing to the inability of many household respondents to identify animals to the species or genus level. Each different type of animal/action by an animal individually listed was counted to generate the total number of 'types of wildlife' observed by each household. This term is not relative to any grouping or category, and instead counts each individual entry written in the survey for each household respondent.

### 2.3. Backyard Animal Activity Surveys

Participants in the backyard camera surveys for wildlife, conducted between May and June 2019, volunteered with written consent after reading the information given in the written surveys. Each yard had four motion-sensor infrared Reconyx Hyperfire PC800 cameras, with two positioned in more 'open' areas that had little canopy or structure cover and no to few plants beyond manicured lawns. The other two were positioned in more 'vegetated' areas, or areas that offered some cover by built structures. Cameras were set to take bursts of 10 images upon being triggered, day and night, for a period of three nights and days. Cameras were secured to a tree or existing structure in yards at a height of ~20 cm above ground, and on a ~10$^{\circ}$ angle to target small to medium-sized terrestrial mammals [58], as well as reptiles and ground-active birds. A scent lure of sponge soaked in fish oil was placed in a 6 cm diameter, 20 cm in length sealed PVC pipe that contained a series of 5 mm diameter holes, and was secured to the ground ~1.5–2 m from each camera. A bait of peanut butter, honey, and oats was also scattered around this. Camera images were scored for species (wildlife and domestic pets), habitat (open/vegetated) and time period (day/night) of activity. Native birds were grouped together owing to the small number of each individual species observed, their common ability to use flight to escape, and their high similarities in active times and habitats. Similarly, brown and black rats were also grouped together, owing to their common urban behaviours [35]. Households with pets that had access to yards were encouraged to allow their pets to access the yards as

they would normally. Repeat images of the same visit by an animal were removed before analyses. This was determined based on the consecutive increases in time, and the size and shape of the animal, as well as the position of the animal being the same or highly similar to the last image.

*2.4. Statistical Analyses*

Effects on the number of types of wildlife observed in yards by householders from the influences of sound, light, garden type, fence type, water resources, pesticide and fertiliser use, and pet ownership and yard access were tested using analysis of similarity (ANOSIM), with 9999 permutations. ANOSIM tests were set with a *gower* dissimilarity distance metric to account for mixed count and ordered factor data, and uneven sample sizes across the categories [59]. Boxplots of the raw data comparing the number of types of wildlife observed under the different categorical response variables were constructed to visualise the direction of any differences. All statistical tests were performed using the statistical software *R*, version 4.0.2 [60]. The ANOSIMs were run using the *vegan* package [59]. Correlations between each of the written survey responses were tested using Pearson's correlations with *P*-values in the package *ggpubr* [61] and were plotted visually using the package *corrplot* [62]. Multinomial logistic regression was run on the video data, to account for the nominal category outcomes in testing the effects of period (day/night) and habitat (open/vegetated) on the animals observed on camera at each of the houses collectively, using the package *nnet* [63]. The grouping of introduced rats (both brown and black rats) was used as the reference animal for comparison, owing to their long history of commensalism with humans [32,35]. Model fit was deemed adequate if McFadden's Pseudo $R^2$ values were between 0.2 and 0.4 [64]. Bar charts of the raw frequencies with which species/groupings of animals were observed by day/night and in open/vegetated habitats were used to visualise the differences relative to the animals and across all groups.

## 3. Results

*3.1. Household Activity Surveys*

Of the 400 houses that were letterbox dropped, 50 households responded by answering all questions (12.5%), and 41 of these observed some form of wildlife activity in their yards. Almost half the respondents owned a pet (*n* = 24), and half of these owned more than one type of pet. Only two of the households that owned pets contained them inside without any yard access, while the others allowed varied pet access to yards, with 38% of these always residing in the yard. Six households allowed their pets to roam freely off their properties; four of these owned cats, another a dog, and the other chickens. From the survey responses only two households retained mostly natural elements in their yards; all remaining yards were modified to varied degrees (Table 1). All properties had fences. Most of these were made of fully enclosed solid materials such as bricks or metal (36%), while the others were open timber or metal material with gaps; half of all the fences (50%) had a gap between the fence and ground (Table 1). More than half the households used a form of fertiliser (66%) and pesticides (54%), a light in the yard at night (56%), and had some form of anthropogenic noise that could be heard in yards (32–60%) (Table 1). All but four households watered their gardens regularly, most did so weekly (*n* = 23), followed by every two months (*n* = 16), and once to twice monthly (*n* = 7) (Table 1). More than a quarter (*n* = 14) of the households had no permanent open water source in their yards (Table 1). Most survey respondents (92%) used the surrounding conservation area and larger green spaces for some form of recreational activity (Table 1).

**Table 1.** A summary of responses by householders to questions pertaining to wildlife in their yards and yard management activities. A total of 50 households responded to all questions in the written survey. Categorisations are based on information received from householders.

| Pets | | | | | | | |
|---|---|---|---|---|---|---|---|
| none | dog | cat | bird | chickens | goat | rabbit | |
| 26 | 17 | 11 | 7 | 6 | 1 | 1 | |

| Garden type | | | | | | | |
|---|---|---|---|---|---|---|---|
| semi-natural | modified-natural | semi-modified | modified | very modified | | | |
| 1 | 1 | 17 | 19 | 12 | | | |

| Watering frequency | | | | | | | |
|---|---|---|---|---|---|---|---|
| never | bimonthly | monthly | fortnightly | once a week | twice a week | second daily | daily |
| 4 | 16 | 3 | 4 | 10 | 7 | 2 | 4 |

| Continual water source available | | | | | | | |
|---|---|---|---|---|---|---|---|
| yes | no | | | | | | |
| 36 | 14 | | | | | | |

| Fertiliser frequency | | | | | | | |
|---|---|---|---|---|---|---|---|
| never | yearly | 6 monthly | 4 monthly | 3 monthly | 2 monthly | monthly | |
| 17 | 8 | 9 | 8 | 6 | 1 | 1 | |

| A combination of fertiliser types was used | | | | | | | |
|---|---|---|---|---|---|---|---|
| compost | fish emulsion/seaweed solution | Manure | blood and bone | wetting agent | targeted fertiliser (plant/lawn/fruit/flower) | dynamic lifter | weed and feed |
| 4 | 5 | 8 | 1 | 1 | 16 | 9 | 2 |

| Pesticide frequency | | | | | | | |
|---|---|---|---|---|---|---|---|
| never | 2 yearly | yearly | 6 monthly | 4 monthly | 3 monthly | 2 monthly | monthly |
| 23 | 3 | 9 | 7 | 5 | 2 | 1 | 1 |

| A combination of pesticide types was used | | | | | | | | |
|---|---|---|---|---|---|---|---|---|
| ant powder/spray | cockroach spray/bait | spider spray | professional multi pesticides | insecticide (pyrethrum/baythroid/bifenthrin) | fruit tree fungicide | lawn weeder (bindi/clover, roundup, weed and feed) | natural pesticide spray | home remedies (white oil, hot water, vinegar, garlic, chilli) |
| 4 | 5 | 2 | 4 | 5 | 2 | 9 | 2 | 9 |

| Light in yard frequency | | | | | | | | |
|---|---|---|---|---|---|---|---|---|
| never | rarely | monthly | weekly | sensor | all night | 6 h night | 3 h night | 1 h night |
| 22 | 5 | 1 | 2 | 15 | 1 | 2 | 1 | 1 |

**Table 1.** *Cont.*

| Light in yard watts | | | | | | |
|---|---|---|---|---|---|---|
| 0 W | 40 W | 50 W | 60 W | 100 W | 150 W | 200 W |
| 22 | 1 | 7 | 12 | 4 | 1 | 3 |

| Appliance/people noise frequency | | | | | | |
|---|---|---|---|---|---|---|
| never | rarely | fortnightly | weekly | daily | afternoons | evenings |
| 20 | 3 | 2 | 1 | 14 | 1 | 8 |

| Moderate music frequency | | | | | |
|---|---|---|---|---|---|
| never | 3 monthly | weekly | daily | mornings | evenings |
| 34 | 1 | 2 | 9 | 2 | 2 |

| Fence type | | | |
|---|---|---|---|
| open gap | full gap | open ground | full ground |
| 14 | 11 | 4 | 21 |

| Primary activity in surrounding conservation area and connected green spaces | | | | | | |
|---|---|---|---|---|---|---|
| none | walk | run | cycle | dog walk | horse ride | mowing |
| 4 | 33 | 4 | 2 | 5 | 1 | 1 |

| Frequency of this activity | | | | | | | |
|---|---|---|---|---|---|---|---|
| none | rarely | 3 monthly | monthly | weekly | twice a week | five times a week | daily |
| 4 | 1 | 1 | 9 | 19 | 4 | 1 | 11 |

| Secondary activities in surrounding conservation area and connected green spaces (multiple for some) | | | |
|---|---|---|---|
| none | walk | cycle | dog walk |
| 31 | 7 | 13 | 2 |

| Frequency of these activities | | | | |
|---|---|---|---|---|
| none | 6 monthly | 4 monthly | monthly | weekly |
| 31 | 1 | 1 | 11 | 6 |

Correlation between the number of types of wildlife observed and the surveyed variables was mostly weak (Figure 1). The category rankings of these variables mean that the number of types of wildlife observed increased when yards were watered more frequently (r = 0.07, P = 0.61), a water source was available (r = 0.23, P = 0.10), the frequency of noise or light (including wattage) increased (r = 0.06, P = 0.70; r = 0.15, P = 0.30; r = 0.16, P = 0.27, respectively), fertiliser application frequency increased (r = 0.08, P = 0.58), pets were allowed in yards more (r = 0.09, P = 0.55), and with the ownership of one or more dogs/chickens/goats (r = 0.17, P = 0.25: r = 0.23, P = 0.12; r = 0.23, P = 0.12, respectively). Contrastingly, the number of types of wildlife observed decreased when gardens were more modified (r = −0.09, P = 0.52), when pesticides were applied more (r = −0.12, P = 0.41), with the ownership of one or more rabbits/birds/cats (r = −0.02, P = 0.62; r = −0.22, P = 0.12; r = −0.11, P = 0.43, respectively), and when there was no gap between ground and fence and the fence material was closed (r = −0.45, P = 0.001), with the latter being the strongest correlation observed and the only one with an associated *P*-value < 0.05. Use of pesticides, fence type, frequency of anthropogenic noise in yards, whether pets had access to yards and the frequency of this, were the only factors that showed variance that was significant within groups as predictors of wildlife in yards more than random chance (Table 2; Figure 2). However, as ANOSIM R values were closer to zero than one, the majority of differences observed were across households more so than within the different categorical groups (Table 2).

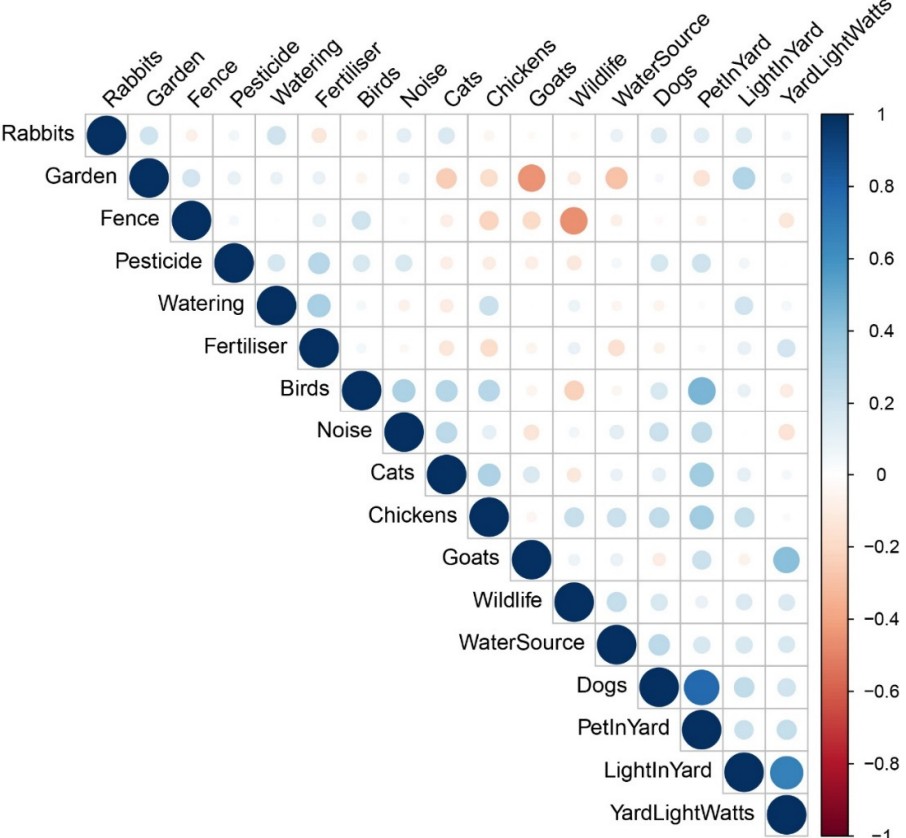

**Figure 1.** Correlation plot for the number of types of wildlife observed (Wildlife) and variables associated with the data from householders who responded to a questionnaire survey. Positive correlations are in blue and negative in red, and the colour intensity and size of the circle are relative to the level of correlation. Animal variables (rabbits, birds, cats, chickens, goats, dogs, and wildlife) are a total count for domestic pets owned and types of wildlife observed. Pet in yard, light in yard, noise, watering, pesticide, and fertiliser are all categories relative to frequencies they are applied to/in yards, with increases relative to increased applications. Garden is categorised relative to modification, ranging from the least to most. Fence is categorised relative to the wildlife access it permits based on

the material and gaps to the ground, ranging from most access to least. Yard light watts is categorised to increase relative to the wattage. Water source is a yes (1) or no (0) category, relative to if an open water source is kept in a yard.

**Table 2.** Analyses of similarity results testing the number of types of wildlife observed at a property against each categorical variable independently.

| Variable | ANOSIM R | Significance |
|---|---|---|
| Yard Light Watts | 0.07164 | 0.1473 |
| Light Used in Yard (y/n) | −0.02777 | 0.8474 |
| Light in Yard Frequency | 0.06116 | 0.1553 |
| Water Source Available (y/n) | −0.09947 | 0.9712 |
| Watering Frequency | −0.04416 | 0.7341 |
| Garden Type | −0.04067 | 0.8452 |
| Fertiliser Used (y/n) | −0.05675 | 0.8682 |
| Fertiliser Frequency | −0.0189 | 0.5983 |
| Pesticide Used (y/n) | 0.05445 | 0.0514 |
| Pesticide Frequency | −0.01336 | 0.5439 |
| Fence Type | 0.08198 | 0.0586 |
| Noise in Yard (y/n) | −0.105 | 0.9609 |
| Noise Frequency | 0.1588 | 0.0118 |
| Pet in Yard (y/n) | 0.07224 | 0.0342 |
| Pet in Yard Frequency | 0.1686 | 0.0148 |
| Number of Cats Present | 0.01956 | 0.3808 |
| Number of Dogs Present | 0.1019 | 0.0812 |

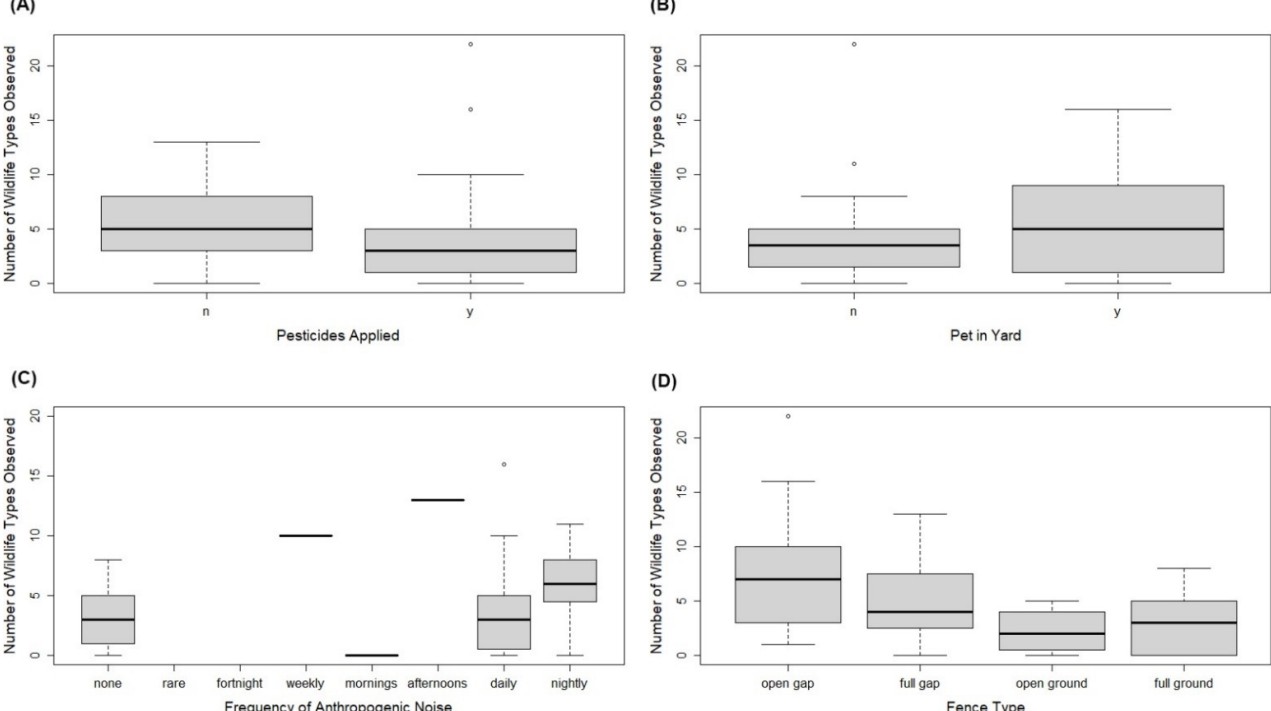

**Figure 2.** Boxplots of raw data for the number of types of wildlife species observed in surveyed householders' yards, by (**A**) pesticides applied, (**B**) pets in yard, (**C**) frequency of anthropogenic noise heard in yard, and (**D**) fence type (*open* or fully enclosed material—*full*) relative to access to wildlife that it permits (based on *gap* between fence and ground or none—*ground*). Each showed significant differences in ANOSIM analyses.

### 3.2. Backyard Animal Activity Surveys

From the survey respondents, 21 (42%) volunteered to have motion-sensor cameras set in their yards to monitor wildlife activity. From this, the number of samples for each yard garden type category was not even (*very modified* = 38%, *modified* = 28%, *semi-modified* = 24%, *modified-natural* = 5%, *semi-natural* = 5%). In total, 13,116 images were collected, and comprised of 333 occasions when animals made visits that each amassed multiple images. Most images were captured at night (70%) and were almost evenly spread across the open (47%) and more vegetated habitats (52%). Animals observed, in order from most to least, were: the common brushtail possum—*Trichosurus vulpecula* (29%); domestic dog—*Canis familiaris* (22%); black and brown rats (grouped together as 'introduced rats' 18%); brown antechinus—*Antechinus stuartii* (13%); buff banded rail—*Gallirallus philippensis*, bush turkey—*Alectura lathemi*, crested pigeon—*Ocyphaps lophotes*, laughing kookaburra—*Dacelo novaeguineae*, Australian magpie—*Gymnorhina tibicen*, noisy miner—*Manorina melanocephala* (all grouped together as 'native birds' 10%); northern brown bandicoot—*Isoodon macrourus* (4%); domestic cat (4%); and red fox (0.3%) (Supplementary Table S1). Most were observed in semi-modified yards (45%) and very modified yards (29%).

Most animals photographed in people's yards were observed more by night than by day (Figure 3), the exceptions being native birds and dogs that were each photographed significantly more by day compared to the night and the relative observations of introduced rats (Table 3). The common brushtail possum, northern brown bandicoot, and red fox were each observed exclusively at night (Figure 3). The brown antechinus, northern brown bandicoot, native birds, and domestic cat were each observed relatively less often than introduced rats in open habitats compared to more vegetated habitats (Table 3, Figure 3). The differences between the brown antechinus and introduced rats in habitat-type use were significant (Table 3). There were overall more observations of native birds, however, in open compared to the more vegetated areas when comparing within this animal group only (Figure 3). Numbers of observations of the introduced rats across habitats were the most similar, but more observations were made in the more vegetated habitats when comparing within this group of animals only (Figure 3). The common brushtail possum, dog, and red fox were each observed relatively more than introduced rats in open compared to the more vegetated habitats (Table 3, Figure 3). However, there were overall more observations of the common brushtail possum in the more vegetated compared to open habitats when comparing within this animal group only (Figure 3).

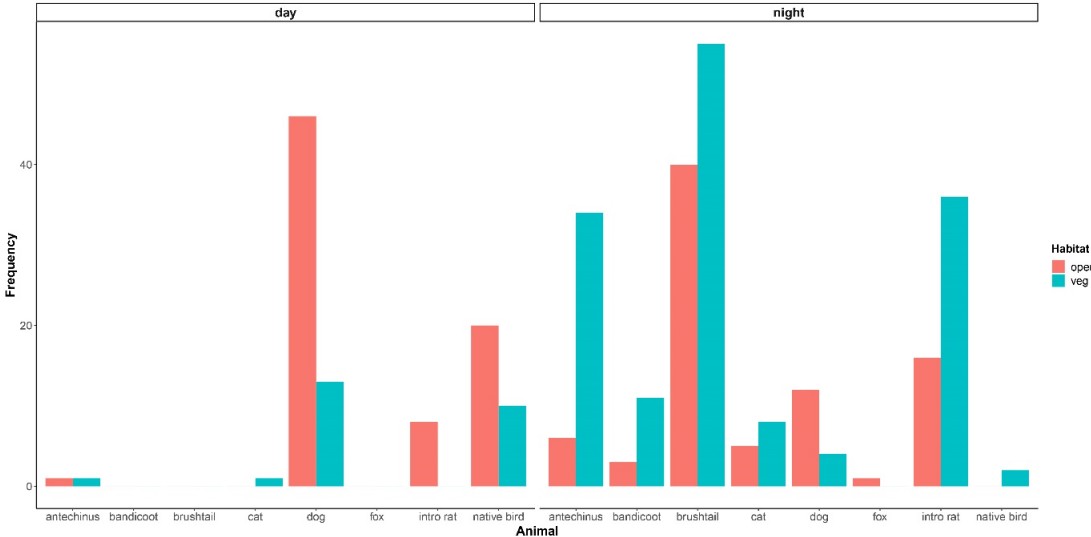

**Figure 3.** Bar chart of the raw frequency of observations for each animal species/group, in open and more vegetated habitats, divided by day and night. Repeat images of single occurrences per animal

are not included. Animals observed were: brown antechinus—*Antechinus stuartii* (antechinus); northern brown bandicoot—*Isoodon macrourus* (bandicoot); common brushtail possum—*Trichosurus vulpecula* (brushtail); buff banded rail—*Gallirallus philippensis*, bush turkey—*Alectura lathemi*, crested pigeon—*Ocyphaps lophotes*, laughing kookaburra—*Dacelo novaeguineae*, Australian magpie—*Gymnorhina tibicen*, noisy miner—*Manorina melanocephala* (all grouped together as 'native bird'); black rat—*Rattus rattus* and brown rat—*R. norvegicus* (grouped together as 'intro rat'); domestic dog—*Canis familiaris* (dog); domestic cat—*Felis catus* (cat); and red fox—*Vulpes vulpes* (fox).

**Table 3.** Multinomial logistic regression results for each animal species/group detected by motion-sensor cameras, analysed against period (day/night) and habitat (open/vegetation cover). The animals observed were: brown antechinus—*Antechinus stuartii* (antechinus); northern brown bandicoot—*Isoodon macrourus* (bandicoot); common brushtail possum—*Trichosurus vulpecula* (brushtail); buff banded rail—*Gallirallus philippensis*, bush turkey—*Alectura lathemi*, crested pigeon—*Ocyphaps lophotes*, laughing kookaburra—*Dacelo novaeguineae,* Australian magpie—*Gymnorhina tibicen,* noisy miner—*Manorina melanocephala* (all grouped together as 'native bird'); black rat—*Rattus rattus* and brown rat—*R. norvegicus* (grouped together as 'introduced rat'); domestic dog—*Canis familiaris* (dog); domestic cat—*Felis catus* (cat); and red fox—*Vulpes vulpes* (fox). All outputs have been exponentiated from the logit scale, except the standard error. Introduced rats were set as the reference level, based on their long history of commensalism with humans.

| Response Variable | Explanatory Variable | Estimate | Standard Error | Test Statistic | P-Value | Confidence Interval (Low) | Confidence Interval (High) |
|---|---|---|---|---|---|---|---|
| antechinus | (Intercept) | 1.01 | 0.24 | 0.05 | 0.96 | 0.63 | 1.62 |
| antechinus | Area (open) | 0.33 | 0.50 | $-2.24$ | 0.03 | 0.12 | 0.87 |
| antechinus | Period (day) | 0.45 | 0.84 | $-0.96$ | 0.34 | 0.09 | 2.31 |
| bandicoot | (Intercept) | 0.33 | 0.35 | $-3.20$ | 0.00 | 0.17 | 0.65 |
| bandicoot | Area (open) | 0.49 | 0.71 | $-1.01$ | 0.31 | 0.12 | 1.95 |
| bandicoot | Period (day) | 0.00 | 0.00 | $-1.96 \times 10^{14}$ | 0.00 | 0.00 | 0.00 |
| brushtail | (Intercept) | 1.65 | 0.22 | 2.30 | 0.02 | 1.08 | 2.52 |
| brushtail | Area (open) | 1.30 | 0.35 | 0.76 | 0.45 | 0.66 | 2.56 |
| brushtail | Period (day) | 0.00 | 0.00 | $-6.59 \times 10^{15}$ | 0.00 | 0.00 | 0.00 |
| cat | (Intercept) | 0.26 | 0.38 | $-3.59$ | 0.00 | 0.12 | 0.54 |
| cat | Area (open) | 0.90 | 0.63 | $-0.17$ | 0.87 | 0.26 | 3.08 |
| cat | Period (day) | 0.52 | 1.12 | $-0.59$ | 0.56 | 0.06 | 4.66 |
| dog | (Intercept) | 0.21 | 0.37 | $-4.23$ | 0.00 | 0.10 | 0.43 |
| dog | Area (open) | 2.38 | 0.46 | 1.88 | 0.06 | 0.97 | 5.87 |
| dog | Period (day) | 18.62 | 0.49 | 6.02 | 0.00 | 7.18 | 48.27 |
| fox | (Intercept) | 0.00 | 0.51 | $-242.65$ | 0.00 | 0.00 | 0.00 |
| fox | Area (open) | $5.38 \times 10^{52}$ | 0.51 | 236.94 | 0.00 | $1.97 \times 10^{52}$ | $1.47 \times 10^{53}$ |
| fox | Period (day) | 0.00 | 56.45 | $-0.12$ | 0.90 | 0.00 | $1.22 \times 10^{45}$ |
| native bird | (Intercept) | 0.04 | 0.75 | $-4.31$ | 0.00 | 0.01 | 0.17 |
| native bird | Area (open) | 0.89 | 0.58 | $-0.19$ | 0.85 | 0.29 | 2.78 |
| native bird | Period (day) | 100.95 | 0.84 | 5.47 | 0.00 | 19.30 | 527.98 |

## 4. Discussion

We found that observations of wildlife in yards by residents were associated with ease of access by fauna under or through fences, reduced pesticide use, increased levels of anthropogenic noise, and increased yard access by pets. Considering the slight increase in the number of types of wildlife observed by householders with supplementary water sources, increased watering frequency, and decreased levels of modification in their yards, we have also investigated these results below, owing to the ease of their application in yard management. Of the households that volunteered to have cameras in their yards, we found that the natural activity periods of native fauna were maintained, and that brown antechinuses, northern brown bandicoots, native birds, and domestic cats used shelter or more vegetated habitats more than open habitats when compared to the commensal introduced rats' activity. Native birds, however, used open areas overall more than the more vegetated or sheltered areas. Similarly, the common brushtail possum, although observed to use open areas comparatively more than the introduced rats, still used more vegetated or

sheltered habitats more than open habitats. The domestic dog and red fox used open areas more than the more vegetated or sheltered habitats, and more than the introduced rats did. As we outline further below, these findings can be used to help motivate householder responsibility for urban wildlife conservation.

Fencing creates a vertical obstacle for wildlife, which may incite stress for animals trying to find an access point through it, particularly if resources are sensed on the other side, and dependent upon the fence material, injuries may occur too [65]. As fences around households are used as property boundaries, for privacy, defense, and to keep and protect pets, they are often of considerable height, so wildlife access to yards is likely mostly where gaps or holes occur underneath. This being the case, it is not surprising that yards with fences that were open or had gaps to the ground were associated with increased numbers of different types of wildlife observed by households, which also supports previous findings [66,67]. Urban residents could be informed of the benefits to wildlife in allowing an ease of transit in and out of their yards via positioning some low gaps between the fence and the ground, at least where residents' needs are not compromised.

Householders may manage their yards with pesticides to meet personal preferences or societal standards, in turn affecting the local urban ecosystem [43,68]. Pesticide use in private yards can negatively impact the wider environment [19] by reducing communities of invertebrates, including non-target soil microfauna [69,70] and flower-visiting insects [71], as well as floristic communities [72]. As this impacts food resources for many types of wildlife, it may reduce the allure of overcoming the obstacles and risks associated with household yards, and hence may be why fewer types of wildlife were observed with higher rates of pesticide application in our study. Certainly, pesticide use has been negatively correlated with occupancy of small mammal species, with flow-on impacts to predators [73]. Routine pesticide use in household yards could be replaced with integrated pest management (IPM) that aims to reduce impacts on the ecosystem by using cultural or mechanical practices and biological control agents over chemicals where possible [74]. Increasing global use of IPM has seen successful moth control in New Zealand [75], mealybug control in vineyards [76], and in private yard maintenance in Canada, where the use of pesticides has been largely banned or restricted [77]. However, for IPM to be successful in urban landscapes, community-wide efforts may be required. While IPM may provide an accessible alternative approach to plant pest management in yards, further research is needed to detail the benefits of allowing spiders, cockroaches, ants, and rodents that are regularly targeted with chemical deterrents to persist in or around yards, or to encourage more environmentally friendly methods to be adopted by households to manage them. Research on biological deterrents for pest management, particularly using olfactory cues to deter invertebrates [78] or small mammals [79], is showing promise and should yield useable products for households.

Increased frequency of anthropogenic noise and yard access by pets in households were each related to more types of wildlife observed by households in our study. While this result seems counterintuitive, owing to each factor being a known stressor to wildlife [20,80], it may reflect the increased observation opportunities that each would create. Anthropogenic noises (music, machinery, appliances, and loud group activities such as sports) occur in or near household yards, thus increasing observation opportunities. Pets often alert owners to wildlife by vocalising or giving chase, also increasing people's chances of making observations. Further, if pets are fed outside then this may act as a lure. As yard light use is often associated with time spent outside at night, wildlife observation opportunities would also increase under these potentially disruptive [22] conditions, although we observed this to a lesser degree than noise and pet in yard frequencies. Motivating management of anthropogenic noise, light, and pet yard activity is challenging in that this would impose on people's lifestyles and habits. Indeed, research on pet cat management has been exploring how to better motivate owners to consider wildlife [21,81], and some government regulations have come from this [21,82]. Similarly, research is increasing on the effects of pet dogs on wildlife too, and how to mitigate them [83]. Still, more research



is required on how to motivate households to manage pet yard access, as well as other wildlife disruptors such as noise and light, whilst not requiring households to significantly compromise their lifestyle.

Regular watering of yards can benefit immediate surrounding areas with run-off, and can provide a contrast to nearby weather-affected environments. These effects may become increasingly apparent, as climate change in many areas is predicted to exacerbate soil moisture deficits via decreased rain and increased temperatures [84]. The benefits of frequent watering to wildlife may include increased food resources as the abundance and distribution of many invertebrate species is associated with adequate moisture levels [85,86]. As water itself is a necessity for animal survival, supplementary available water sources in yards may also be an attractant to enter them from drier surroundings. These benefits may account for the positive correlation between numbers of wildlife species and watering frequency, or open water availability. Government, educational, and non-government organisation initiatives that promote gardening for wildlife all suggest that an open available water source is necessary for provisioning wildlife, along with food and shelter [87]. Our results of increased types of wildlife observed in yards with increased water availability supports this suggestion.

Decreased modification of yards allows more opportunity for wildlife to display natural behaviours. Higher abundances of mammals are often associated with increased natural vegetation in yards [88,89], and our results are also consistent with this. While we did not quantify native vegetation directly in our study, people who described their yards as having fewer modified features also observed more types of wildlife. Further, although *very modified yards* were surveyed more by camera, there were higher numbers of wild animal visitors captured at the second most surveyed yard type—*semi-modified*, which were comparatively more natural. There are benefits to human health associated with connecting to nature [90–92]. Trees are also of benefit as they mitigate climate by reducing cooling or heating needs [93–95]. As such, these benefits could be used in concert with the benefits to sustaining urban biodiversity, to motivate householders to maintain high levels of naturalness in their yards.

It was evident from our camera images that both open and more vegetated/covered habitats were used by wildlife in people's yards. However, the more vegetated areas were used more, except by dogs, native birds, and the one red fox observed. The use by native small mammals of the more vegetated habitats supports the need for household yards to maintain some dense vegetated patches amongst some open patches, to facilitate stress-reducing behaviours such as hiding or escaping [29]. Given that patchworks of vegetation may be visually appealing for people too [96], there is a duality in what could drive motivation to use wildlife-friendly gardening. One mammal, the common brushtail possum, used open areas more than the commensal rats did, perhaps owing to the bold personalities and problem-solving abilities of this possum [97], making it successful in urban settings, and to its arboreal ability that opens up additional quick escape routes. As the common brushtail possum was the most frequently observed species, this may attest to its traits making it a successful urban adapter [98]. The native birds observed also preferred open areas and by their use of flight to escape threats are common urban adapters/exploiters [98]. Domestic cats were observed more in vegetated habitats where small mammals were also most frequent, which supports the need for wildlife-friendly pet cat management, such as containment or use of pet cat pens [21]. Northern brown bandicoots were the least frequently observed native animal in household yards, despite being one of the most common species in the surrounding green spaces in our companion studies [27,28]. As a large portion of their diet is invertebrates, bandicoot presence may be attributed to the benefits of additional watering [86], and their reduced presence may be attributed to the negative impacts of pesticide use [70]. Given their strictly terrestrial movements and relatively larger size, northern brown bandicoots may also have been inhibited by access to some yards, further supporting the management of fences to incorporate some level of gap between the ground and fence.

## 5. Conclusions

With the continued growth of the human population, yard management decisions of individual households will collectively affect urban ecology and biodiversity. As such, urban wildlife conservation needs to empower people to make educated decisions about how their actions and structures may affect wildlife [40,99]. Our study provides evidence that wildlife use household yards and that human activities and yard management affect this, supporting the need for environmentally-friendly gardening. The challenge, however, is in motivating householders to do this. We have provided some information indicating the positive and negative effects on wildlife that different yard management practices can have. However, there is evidence that direct environmental experience, such as citizen science, volunteering on research projects, or guided nature tours, can increase the likelihood of such information being used to start wildlife-friendly gardening, and of the practices being retained over time [71]. Almost all the households that responded to our written survey made regular use of the large state conservation area surrounding their properties, and this may suggest support for the idea that connecting with nature fosters a level of responsibility [41]. Whilst wildlife occurred in and around household yards despite the attendant stressors from human activities, it is important to note that these activities may have negative impacts that go unseen by householders, in turn impacting local biodiversity. Further research into how best to motivate householders to be considerate neighbours to surrounding wildlife will help to link the science with action.

**Supplementary Materials:** The following supporting information can be downloaded at: https://www.mdpi.com/article/10.3390/d14040263/s1. Supplementary Table S1: A summary of the animals captured on remote sensor camera traps in the backyards of householders who volunteered for the camera survey, across day and night, has been made available with this publication. Supplementary Information: The written survey questions mailed to residents in the study area.

**Author Contributions:** L.L.F., C.R.D. and C.R.P. conceived this study. L.L.F. performed the research and statistical analyses. L.L.F. wrote the initial manuscript draft, and all authors edited and contributed to the subsequent drafts. All authors have read and agreed to the published version of the manuscript.

**Funding:** This research was supported by a Holsworth Wildlife Research Endowment, and a Lake Macquarie Environmental Research Grant funded by Lake Macquarie City Council and other sponsors, which in 2018–2019 included Hunter Water Corporation, Delta Electricity and Origin Energy. L.L.F. was supported by an Australian Government Research Training Program Stipend, and a University of Sydney Merit Award Scholarship.

**Institutional Review Board Statement:** The animal study protocol was approved by the Ethics Committee of the University of Sydney (protocol code: 2017/1275, date of approval: 6 December 2017), and was performed in the area under a New South Wales Scientific License (SL102024). The study was also conducted in accordance with the Declaration of Helsinki and approved by the Ethics Committee of the University of Sydney (protocol code: 2017/977, date of approval 21 December 2017).

**Data Availability Statement:** Raw data, protecting the identity of household owners, may be made available upon email request to L.L.F., only if participant approval is given.

**Acknowledgments:** We acknowledge the traditional owners of country throughout Australia and recognise their continuing connection to lands, waters and communities. We pay our respect to Aboriginal and Torres Strait Islander cultures; and to Elders past, present and emerging. We recognise and acknowledge the Aboriginal people, known today as the Awabakal, as the traditional Custodians of the land presently known as the suburbs of Whitebridge and Dudley. We are grateful to all volunteers that participated in this study. We are grateful to John Clulow for stimulating discussions on urban wildlife in the Lake Macquarie area and for his direction to sites of high wildlife activity. We are also grateful to Bobby Tamayo for his endless support and direction both logistically and in creative problem solving for in-field research.

**Conflicts of Interest:** The authors declare that the research was conducted in the absence of any commercial or financial relationships that could be construed as a potential conflict of interest.

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
