# Peer review of "Backyard Biomes: Is Anyone There? Improving Public Awareness of Urban Wildlife Activity"

_diversity, doi:10.3390/d14040263_

Round 1

Reviewer 1 Report

Review of the paper „Backyard biomes: is anyone there? Improving public awareness of urban wildlife activity”

The manuscript deals with the issue of public awareness of wildlife, which affects the protection of nature in an era of rapidly increasing urbanization and the loss of habitats for wildlife. The topic is interesting, but the manuscript needs to be improved. Below are some suggestions.

Abstract. In general, a bit too long – suggest removing 2 to 3 sentences. L12 – it is unclear who or what is affected by disturbances and benefits. L14 – in this place, it is not clear the meaning of "type" written in quotation marks. I suggest using the word group or category if the use of the quotation marks indicates that it is not a taxonomy type, because abstract readers do not know what it is. L16 – rather than "more type of wildlife" I suggest wildlife diversity or richness. L20 – overgrown rather than vegetated; open areas also can contain plant communities. L27 – “sheltered or vegetated habitats over open habitats” – not clear (see above). L28-31 – please rephrase, readers will not understand within which group and what is the matter with habitat types. The abstract lacks conclusions from the conducted research and conservation implementations.

Introduction. L43 – please, choose whether globally or in many areas. L61-80 – for the purposes of the statements in this paragraph, there is a need to inform readers what open and other areas are. I’m not sure the authors understand open areas as concrete squares with nothing. Even in the open air, it seems to me that there is vegetation, if only in the sense of grass and soil organisms. So the reader doesn’t really know where the borders between open and the rest of the areas are. L81-92 – I do not understand the meaning of this paragraph – starts with the significance of urban complexity, then contains information on the role of city dwellers in nature conservation, and finally information on the problem of not considering people's perceptions in nature conservation. I do not understand the significance of this paragraph in the article on the role of courtyards in nature protection. It must be better linked to the message of the paper. L93-107 – there is no information on private property in this paragraph. It is difficult for the government to tell someone how to manage private property. There are, however, rules on permits to removing trees from private land, rules governing land management with the need to take into account local utilities, such as power lines, and landscape laws. The question is what percentage of urban areas is owned by private people – probably relatively few. These are issues that would be worth considering in the introduction or in the discussion. L134 – please, explain the activities and whether they apply to courtyards or urban areas in general. L134-135 – why is the rats’ comparison so highlighted in the manuscript questions? Predictions are not closely related to the questions asked to conduct the research.

Materials and Methods. A map of the study area and its location in a larger space seems to be useful. L161 – modification and naturalness should be explained here. L169-190 – I suggest organizing this paragraph in a table with the criteria for matching specific areas to the concepts of the natural scale. Statistical analysis – you need to write what version of R you used, preferably by placing the information at the beginning or end of the chapter, because not only one test was prepared in R. 

Results. The correlations, at least selected, could be shown in charts. Figure 1. – this plot was made in the corrplot package, which is not written in statistical analyses. Figure 3. – I encourage you to reflect once again on the concepts of open and vegetated and apply them throughout the manuscript. Was the lawn open or vegetated? Overall, I consider the presentation of the results to be the best part of the manuscript - they are logically arranged, simple enough and at the same time attractively analysed.

Discussion. There is no comment on the low percentage of people who answered the authors’ questions. Does this not mean that the presented results come from a part of society aware of the presence of wildlife in urban areas? L377-378 – not to contain pets, but rather to keep or protect them. L458-483 – very interesting paragraph. Conclusions are reasonable.

The method of citing articles should be adapted to the editorial requirements of the journal. I have read the supplementary table and it is interesting for readers, it can easily be placed as an Appendix in the main text.

Author Response

Reviewer 1

 Dear Reviewer 1,

Thank you kindly for taking the time to review our manuscript. Please find below responses (shown in plain text) to each of your suggested changes / comments (shown in bold text).  

Abstract.

In general, a bit too long – suggest removing 2 to 3 sentences.

The abstract has been edited to reduce the word count, as suggested.

L12 – it is unclear who or what is affected by disturbances and benefits.

In response to the comment regarding L12 of the Abstract, no adjustments have been made. As detailed in lines 1 to 8 of the Abstract, this study focuses on ‘types’ of wildlife, as the data largely come from a written survey response from household respondents that may or may not have accurately identified wildlife species but have differentiated between what they have observed, as they list several animal species/groups. The responses were not all consistent between households and could be unreliable if taken to a species level, so we chose to group these as the number of types of wildlife observed by the householder. Those animals observed on the cameras support this approach to a degree, but do not show all animals listed as observed by household respondents. It was our intention in writing this study to frame it as wildlife types, instead of being more specific, to focus instead on the fact that there may be stressors relative to households affecting wildlife use of yards, from an unbiased human perspective.

L14 – in this place, it is not clear the meaning of "type" written in quotation marks. I suggest using the word group or category if the use of the quotation marks indicates that it is not a taxonomy type, because abstract readers do not know what it is.

In response to the comment regarding L14 of the Abstract, no adjustments have been made in this section. Following the justification given above, the word “type” is used as it is from the householder’s perspective and is not relevant to taxonomy. To further support this, taxonomy is not mentioned in the manuscript. “Group” or “category” would not accurately replace “type” as animals were not counted as groups/categories but rather as each animal (type) that was written in the survey response, be it a species or a group. For example, in a survey response the laughing kookaburra may have been detailed but Australian native birds may also have been detailed, and this response would be counted as 2 types of different wildlife observed. Owing to the word constraints on the Abstract section, the use of the term 'different types of wildlife' is further detailed in the Materials and Methods section and has been edited to make this point clearer, at lines 218-224 of the edited version of the manuscript.

L16 – rather than "more type of wildlife" I suggest wildlife diversity or richness.

In response to the comment regarding L16 of the Abstract, no adjustments have been made. Following the justifications given above and, as diversity or richness were not measured by the household respondents, it would not be accurate or possible to use these terms based on the observations or our respondents alone.

L20 – overgrown rather than vegetated; open areas also can contain plant communities. L27 – “sheltered or vegetated habitats over open habitats” – not clear (see above).

In response to the comment regarding L20 and 27 of the Abstract, no adjustments have been made to this section. The vegetated areas were not necessarily overgrown, but they contained more vegetation than the open areas that had no canopy cover but could have had some grass or minimal low-lying plants. Owing to the word constraints on the Abstract section, the use of the terms ‘open’ and 'vegetated' have been amended and are more explicitly detailed in the Materials and Methods section, at lines 229-231 of the edited version of the manuscript.

L28-31 – please rephrase, readers will not understand within which group and what is the matter with habitat types. The abstract lacks conclusions from the conducted research and conservation implementations.

In response to the comment regarding L28-31 of the Abstract, limited adjustments have been made. As the prior sentences list the species recorded it is clear which species the concluding statement is referring to, without the need to repeat them and unnecessarily increase the word count. Regarding the conservation implications, these may be taken from our results, but as we discuss them relative to each potential disturbance to wildlife, it would be too discursive to repeat all these in the abstract, so instead we provide our general conclusions and note that detailed insights are discussed in the body of the manuscript.

Introduction

L43 – please, choose whether globally or in many areas.

In response to the comment regarding L43 of the Introduction (line 60 of the edited version of the manuscript), the term “many areas” has been removed.

L61-80 – for the purposes of the statements in this paragraph, there is a need to inform readers what open and other areas are. I’m not sure the authors understand open areas as concrete squares with nothing. Even in the open air, it seems to me that there is vegetation, if only in the sense of grass and soil organisms. So the reader doesn’t really know where the borders between open and the rest of the areas are.

In response to the comment regarding L61-80 of the Introduction, open yard areas have been more clearly defined, at lines 86-87 of the edited version of the manuscript.

L81-92 – I do not understand the meaning of this paragraph – starts with the significance of urban complexity, then contains information on the role of city dwellers in nature conservation, and finally information on the problem of not considering people's perceptions in nature conservation. I do not understand the significance of this paragraph in the article on the role of courtyards in nature protection. It must be better linked to the message of the paper.

In response to the comment regarding L81-92 of the Introduction, we argue that this paragraph (lines 103-114 of the edited version of the manuscript) is integral to defining the perspective of this study. That is, people and the way they manage their yards have a strong influence on the types of wildlife that occur in urban environments, and our survey of householders sought to show this. This introductory paragraph details the importance of considering the human element in urban wildlife conservation, and how people's responses may be considered from a management perspective. We amplify this further in the Discussion.

L93-107 – there is no information on private property in this paragraph. It is difficult for the government to tell someone how to manage private property. There are, however, rules on permits to removing trees from private land, rules governing land management with the need to take into account local utilities, such as power lines, and landscape laws. The question is what percentage of urban areas is owned by private people – probably relatively few. These are issues that would be worth considering in the introduction or in the discussion.

In response to the comment regarding L93-107 of the Introduction, we clarify that this paragraph is indeed discussing private property in referring to household yards. We have amended the opening sentence (lines 115-116 of the edited version of the manuscript) to include your suggestions regarding rules governing land management.

L134 – please, explain the activities and whether they apply to courtyards or urban areas in general.

In response to the comment regarding L134 of the Introduction, we note that the different kinds of human activities have been detailed throughout the introduction and are referred to as resources or stressors. To clarify this further, we have however changed the wording at the first mention of such resources and stressors, at lines 75-79 (of the edited version of the manuscript) to make it clear that they are relative to human activities. We have also added the term “as resources or stressors” into the section that details the study questions, at lines 154 and 158 (of the edited version of the manuscript). As the activities measured are given in detail in the Materials and Methods section, at lines 197-224 (of the edited version of the manuscript) where the topics of the survey questions are given, we felt no need to provide further information in the Introduction at the cost of increasing the word count.

L134-135 – why is the rats’ comparison so highlighted in the manuscript questions? Predictions are not closely related to the questions asked to conduct the research.

In response to the comment regarding L134-135 of the Introduction, the comparison to introduced rats is now clarified further. As black and brown rats have been commensal with humans for a long time, they offer consummate examples of animals that persist around human disturbances and have had more time to develop responses than any of the native species observed. The comparison with rats, we believe, therefore offers some insight into how native animals may be coping with urban stressors. More details have been added at lines 95-102 (of the edited version of the manuscript) of the Introduction.

Materials and Methods.

A map of the study area and its location in a larger space seems to be useful.

In response to the request for a map of the study area, one has not been provided, in the interest of protecting the identity of the households asked to participate in this study. In its place we have given a detailed description of the area in the appropriate Materials and Methods section, and we also give reference to a map of the wider area that is provided in our companion study’s publication.

L161 – modification and naturalness should be explained here.

In response to the comment regarding L161 of the Materials and Methods section, an explanation of each modification and naturalness has been added to lines 201-205 (of the edited version of the manuscript).

L169-190 – I suggest organizing this paragraph in a table with the criteria for matching specific areas to the concepts of the natural scale.

In response to the comment regarding L161 of the Materials and Methods section, no action has been taken in the interest of reducing graphical displays in the manuscript. Further, given the context of the information, we feel that it is easier to understand in the written format given.

Statistical analysis – you need to write what version of R you used, preferably by placing the information at the beginning or end of the chapter, because not only one test was prepared in R. 

In response to the comment regarding the statistical analyses details in the Materials and Methods section, the language has been amended and the version of R has been added to line 259 (of the edited version of the manuscript).

Results

The correlations, at least selected, could be shown in charts.

In response to the suggestion that the correlation details given in text be presented in a chart format, no further revision has been undertaken as a succinct summary of the correlation results is shown already in Figure 1. Given the context of the information, of the direction of each variable relative to what the correlation means, only the main correlation results have been highlighted in the text.

Figure 1. – this plot was made in the corrplot package, which is not written in statistical analyses.

In response to the comment regarding Figure 1 of the Results section, the corrplot package has now been cited in the Materials and Methods section, at line 262 (of the edited version of the manuscript).

Figure 3. – I encourage you to reflect once again on the concepts of open and vegetated and apply them throughout the manuscript. Was the lawn open or vegetated?

In response to the comment regarding Figure 3, and the need to detail the concepts of open and vegetated areas, more explicit details have now been given in the Materials and Methods section, as detailed above. The term “more” has also been placed in front of “vegetated” in the Figure 3 caption, and where appropriate in the Results and Discussion sections, to better clarify the differences between the concepts of open and vegetated as we have used them.

Discussion

There is no comment on the low percentage of people who answered the authors’ questions. Does this not mean that the presented results come from a part of society aware of the presence of wildlife in urban areas?

In response to the comment regarding the low number of participant respondents not being presented in the Discussion section, we argue that without knowing the perspectives of the residents that did not respond, it would be naive to presume that all who responded are aware/considerate of wildlife in the area. It is possible they have some interest in this, but it is also possible that they chose to respond for other reasons too. Whether the number of responses could be perceived as low is subjective and needs to consider other factors that may influence responding, and similar study response rates. Without clearly knowing the motivation for responding and not responding, we chose not to discuss this further, to avoid any bias or assumptions. However, this issue is important and we do raise it in a slightly different context and suggest in the Conclusion, lines 557-560 (of the edited version of the manuscript), that the high rate of use of the surrounding nature reserves by the respondents may indicate that their connection with nature, via their indicated frequency of visits to it, may indicate a level of environmental consideration.

L377-378 – not to contain pets, but rather to keep or protect them.

In response to the comment regarding L377-378 of the Discussion, the language has been changed to replace “contain” with “keep and protect”, now on line 432-433 (of the edited version of the manuscript).

L458-483 – very interesting paragraph. Conclusions are reasonable.

The method of citing articles should be adapted to the editorial requirements of the journal.

In response to the comment regarding the reference methods, the reference list in and text methods for citing articles have now been changed to meet the editorial requirements of the journal.

Reviewer 2 Report

  1. The introduction is very large and it should be shortened.
  2. The printed written survey is not included in the Supplemental Information.
  3. How many yards were there in which Reconyx Hyperfire cameras were installed? However, further on the manuscript, these data are available.
  4. How many answers were given in the questionnaires and how many questionnaires were there for statistical analysis?
  5. Explain the purpose of the bait. It seems to me that bait is not needed in such studies. It serves to attract animals. However, why attract them to residential courtyards? Wild animals should not be specially lured to human homes. This creates problems not only for animals, but also for humans. After all, wild animals can bring with them different zoonoses.
  6. Judging by the research results, we come to disappointing conclusions based on the methodology of the study itself. In my opinion, there is very little data for any statistical analysis. Only 50 data have been studied for analysis. If there are no answers to the questionnaire, then the authors could independently go through the households and record the results. Or the authors could have conducted another survey.
  7. A total of 21 yards garden were equipped with cameras to survey the activity of wild animals. It is also not enough to get statistically correct data.
  8. Conclusion. It seems to me that it is quite difficult to get homeowners to make their yards so that wild animals feel good in them. We need to redo these phrases.
  9. Reference and links to publications are not made according to the rules of the journal.

Author Response

Reviewer 2

 Dear Reviewer 2,

Thank you kindly for taking the time to review our manuscript. Please find below responses (shown in plain text) to each of your suggested changes / comments (shown in bold text).  

  1. The introduction is very large and it should be shortened.

In response to the concern regarding the length of the Introduction, it is currently 1141 words, having been reduced somewhat by responses to other reviewer comments and by the replacement of author names and dates by numbers to accord with journal style. We counter that all content now within this section is required to build a case for the need to conserve urban environments, and to do so by including the public and their activities in management designs. This allows us to build the narrative of the underlying conflict between urban wildlife and people.

  1. The printed written survey is not included in the Supplemental Information.

In response to the comment regarding the missing written survey document, it has now been provided as Supplemental Information.

  1. How many yards were there in which Reconyx Hyperfire cameras were installed? However, further on the manuscript, these data are available.

In response to the comment regarding the number of yards surveyed, this information is placed in the results section, at line 353 (of the edited version of the manuscript), as it is a result from the survey mailout that included the study request. The Reconyx Hyperfire model (PC800) has been added to the Materials and Methods section, at line 228 (of the edited version of the manuscript).

  1. How many answers were given in the questionnaires and how many questionnaires were there for statistical analysis?

In response to the comment regarding the number of answers and questionnaire responses used in the statistical analysis, this information has now been provided at the start of the Results section. Further, it is detailed within the Methods section, from line 197 (of the edited version of the manuscript), that: Answers were open ended without scaled options, to get a better understanding of householders’ yard structures and activities without the possible bias from pre-defined categories. It then goes on to detail how responses were categorised. All questions were answered in each of the survey responses, which has now been made clear on 276-277 (of the edited version of the manuscript) of the Results section. 

  1. Explain the purpose of the bait. It seems to me that bait is not needed in such studies. It serves to attract animals. However, why attract them to residential courtyards? Wild animals should not be specially lured to human homes. This creates problems not only for animals, but also for humans. After all, wild animals can bring with them different zoonoses.

In response to the comment on the use of bait, we agree that it is not ideal to lure wildlife into residential yards. However, we believe that given the small amount of bait used, and the availability of supplementary resources in the area that it does not introduce an additional resource too far removed from what wildlife may already find there. Further, given that natural resources are available in the area, it reduces the appeal of the scent lure from further distances than the immediate area. The use of bait lures on cameras is supported, as the infrared light and high frequency sound are detectable by many animals, which may cause them to avoid them, as would neophobic tendencies. The bait simply ensures that animals are in the focal area for long enough for a camera to reliably capture their presence, and to do so in a minimally invasive way.

  1. Judging by the research results, we come to disappointing conclusions based on the methodology of the study itself. In my opinion, there is very little data for any statistical analysis. Only 50 data have been studied for analysis. If there are no answers to the questionnaire, then the authors could independently go through the households and record the results. Or the authors could have conducted another survey.

In response to the comment on the number of survey responses, and suggestions to gain further data, we offer a defence of why we believe our data set is sufficient for analyses and publication. Given the questions asked in the survey, listed in lines 186-195 (of the edited version of the manuscript) of the Materials and Methods section, it would not be possible to answer the questions on behalf of the households that did not respond, as this information is not publicly available. As conducting another survey would require surveying in a location out of the study area, we chose not to do this, so as to keep the influencing factors as similar as possible.  Further, whether the number of responses could be perceived as low is somewhat subjective and needs to consider other factors that may influence responding/not responding, and similar study response rates, which was not possible in the scope of this study. We acknowledge that 50 responses given the number of variables and different influences per household, is not ideal. However, all 50 households answered all questions in our questionnaire, as we now specify in the first line of the results. Moreover, the details of the statistical analyses and results are explicit and justified based on the data, and clearly allow the reader to consider the smaller sample size when interpreting the results. As this is an under-published topic, most likely owing to the requirement for gaining public responses, we feel it important to publish any data gathered on this topic - especially when, as in our case, it is clearly amenable to statistical interpretation - so that it may help to better direct and encourage future studies on this topic.

  1. A total of 21 yards garden were equipped with cameras to survey the activity of wild animals. It is also not enough to get statistically correct data.

In response to the comment on the number of yards that were surveyed with cameras, we offer the same defence as given above, and re-iterate that all data were analysed correctly considering the smaller sample size. Further, all interpretations of the data are given considering them as broad or likely implications of the smaller data set. We note that in many analyses n = 6 is considered a sufficient sample size; our sample size here was 3½-fold greater than that suggested by the rule of thumb that n ≥ 6 is a sufficient minimum.

  1. It seems to me that it is quite difficult to get homeowners to make their yards so that wild animals feel good in them. We need to redo these phrases.

In response to the comment on the Conclusion, in terms of it being difficult to get homeowners to make their yards more attractive to and safer for wild animals, we offer many suggestions on how households could be educated to reduce the unseen conflict between urban wildlife and humans, in the interest of conserving wildlife biodiversity, and recognise the challenges of gaining householder support. Further, we reiterate that this is something that needs to be discussed and considered by people who are arguably becoming more environmentally conscious given the current state of the world.

  1. Reference and links to publications are not made according to the rules of the journal.

In response to the comment regarding the reference methods, the reference list in and text methods for citing articles have now been changed to meet the editorial requirements of the journal.

Reviewer 3 Report

Thank you for this opportunity to review the paper, I greatly enjoyed reading it! The research question and methods are clearly described, and the analysis includes visualizations that make the results easy to understand. I include very minor (and optional) changes in the file attached here.

I believe the body of the paper indicates that the survey questions will be included in the supplementary information, but I did not see that and was interested in it. I also wondered if people indicated a dislike of animals (especially rats) in their yards, and if so, where/how that was expressed.

An ingenious study that provides information that we increasingly need as humans and wildlife live in close proximity. Thank you for your efforts.

Author Response

Reviewer 3

Dear Reviewer 3,

Thank you kindly for taking the time to review our manuscript, we greatly appreciate your kind words and support. Please fid below responses (given in plain text) to each of your suggested edits including those made in text (given in bold).

Thank you for this opportunity to review the paper, I greatly enjoyed reading it! The research question and methods are clearly described, and the analysis includes visualizations that make the results easy to understand. I include very minor (and optional) changes in the file attached here.

In response to the in text suggest grammatical edits, most have been made, and we thank you kindly for the suggestions.

Regarding the comments in text in the Abstract section, to clarify, we are assessing the benefits and deficits to wildlife, and five sightings of one type of animal would indeed count as one.

I believe the body of the paper indicates that the survey questions will be included in the supplementary information, but I did not see that and was interested in it.

In response to the comment, to include the survey questions in the Supplementary Material, this has now been done.

I also wondered if people indicated a dislike of animals (especially rats) in their yards, and if so, where/how that was expressed.

In response to the query of whether people expressed a dislike for rats in their yards, no likes or dislikes of any animals were expressed by any of the written survey respondents.

An ingenious study that provides information that we increasingly need as humans and wildlife live in close proximity. Thank you for your efforts.